# HARP2 Pre-Launch Calibration: Dealing with polarization effects of a Wide Field of View

Noah Sienkiewicz[1], J. Vanderlei Martins[1,2], Brent A. Mcbride[1,2], Xiaoguang Xu[1,2], Anin Puthukkudy[1,2], Rachel Smith[1,2], Roberto Fernandez-Borda[1,2]

[1]Department of Physics, University of Maryland Baltimore County, Baltimore, 21228, USA
[2]Earth and Space Institute, University of Maryland Baltimore County, Maryland, USA

*Correspondence to*: Noah Sienkiewicz (noahs3@email.com)

**Abstract.** The HyperAngular Rainbow Polarimeter (HARP2) is a wide field-of-view (FOV) polarimeter built for the NASA Plankton Aerosol Cloud and Ocean Ecosystem (PACE) mission launched in early 2024. HARP2 measures the linear Stokes
parameters across a 114° x 100° (along-track by cross-track) FOV. In the Fall of 2022, HARP2 underwent calibration at NASA Goddard Space Flight Center (GSFC) Calibration Laboratory (Code 618). HARP2 was characterized for radiometric and polarimetric response across its FOV. We have used telecentric calibration methodology on prior iterations of HARP that involved the normalization of pixels across the FOV such that calibration parameters determined at the center of the charged coupled device (CCD) detector can be used across the entire scene. By using a dual-axis yaw/pitch motorized mount, we
devised two scan patterns to evaluate this methodology for HARP2. The results show that pure intensity measurements do indeed vary minimally across the FOV and therefore can utilize the flat-field normalization (telecentric) technique. On the other hand, images of polarized targets change significantly across the FOV, and calibration parameters determined at the center of the detector used in the wide FOV perform significantly worse than calibration parameters determined at or near to the location of the test (up to 5% mean absolute uncertainty in degree of linear polarization, DoLP). We evaluated the use of
a paraboloid fit of the polarized calibration parameters, at discrete FOV locations, to determine those parameters at a pixel-level resolution. According to the wide FOV results, this process shows a marked improvement for fully polarized (DoLP = 1) calibration data to less than 1% uncertainty after using the paraboloid fit. These results are important for the development of any wide FOV polarimeter, especially those like HARP2 which use a front lens which causes significant barrel distortion and a division of amplitude central optical element leveraging multiple reflections. Full characterization of the source of these
optical effects remains a part of future work but the improved methodology over the telecentric method is currently being implemented in the HARP2 L1B calibration pipeline pending internal review of the implementation in the HARP Image Processing Pipeline.

## 1 Introduction

Spaceborne remote sensing platforms are an indispensable tool for understanding the evolution of the global climate
system as they allow regular coverage of wide swaths of the planet. Multi-angle polarimeters (MAPs) represent a leap forward

in this regard by capturing the full linear Stokes parameters as compared to typical radiometers which measure only the first parameter: total radiance (or reflectance) at the top of the atmosphere (TOA). Clouds, aerosols, and surface targets exhibit distinct qualities in polarization which are emphasized by measurements at multiple view angles. Data from MAPs can characterize cloud droplet size distributions (McBride et al., 2020; Miller et al., 2018) and thermodynamic phase (J. V. Martins

et al., 2011) by leveraging information in the angular distribution of the cloud polarization signal. The retrieval of aerosol properties such as the sphericity of particles (Dubovik et al., 2006), size, and refractive index (Mishchenko & Travis, 1997; Puthukkudy et al., 2020) are also better constrained by MAPs thanks to the additional information encoded in their polarized phase functions. This also leads to stronger aerosol speciation (Hamill et al., 2020) which is important for tracking aerosols sources and their impacts on human health and global climate. Further, aerosol retrievals using MAP data are more accurate

over complex land (Hasekamp & Landgraf, 2007) and ocean surfaces (Hasekamp & Landgraf, 2005), which may present trouble to purely spectral methods.

Different MAP instruments have been designed and tested, often via aircraft campaigns (Dubovik et al., 2018), but all polarimeters take advantage of the fact that direct sunlight enters the atmosphere unpolarized; sometimes this is also described as being uniformly (or randomly) polarized in all directions. When that light impinges upon a particle in the

atmosphere, or an object on the Earth's surface, its state is changed to that of partial polarization, leaving a marker of that interaction that an MAP can capture in a measurement of the linear Stokes parameters: $I$, $Q$, or $U$. For example, reflection off of the ocean may strongly polarize the unpolarized solar signal under certain geometry (Harmel & Chami, 2013). Aloft aerosols may produce different polarization effects (Li et al., 2019). There is no single, optimal MAP instrument design: some designs have utilized multiple telescopes, wide field-of-view (FOV) lenses, and mobile gimbals in the past to achieve a variety of

multi-angle sampling characteristics. To measure polarization, these systems use some internal optical element for the near-simultaneous imaging necessary for the recreation of the linear Stokes vector (Tyo et al., 2006). Natural targets rarely impart the circular polarization (Hansen & Travis, 1974) and therefore it is typically ignored to simplify instrument design and calibration.

Among the modern MAPs is the HyperAngular Rainbow Polarimeter (HARP2), which launched on the NASA

Plankton Aerosol and Cloud Ecosystem (PACE) mission in early 2024 (Remer, Davis, et al., 2019; Remer, Knobelspiesse, et al., 2019) alongside the Spectro-Polarimeter for Planetary Exploration (SPEXone) (Hasekamp et al., 2019). Both have flown aircraft versions on precursor campaigns, such as the Aerosol Characterization from Polarimeter and LIDAR (ACEPOL) campaign and shown good agreement and strong capability in aerosol and cloud retrievals (Fu et al., 2020; Mcbride et al., 2024; Puthukkudy et al., 2020). The goal of PACE is to advance the study of the Earth's land-ocean-atmosphere ecosystem

using, in part, polarimetry (C. R. McClain, 2009; Werdell et al., 2019). To help accomplish this task, MAPs on PACE were asked to meet an 0.5% absolute accuracy in degree of linear polarization (DOLP) (Remer & Boss, 2018). Prior studies of data from the Polarization and Anisotropy of Reflectances for Atmospheric Sciences coupled with Observations from Lidar (PARASOL) satellite, which contained the Polarization and Directionality of the Earth's Reflectances (POLDER) polarimeter (Knobelspiesse et al., 2012), and simulations (Mishchenko et al., 2004) have shown that this level of accuracy allows for

confident estimation of aerosol radiative forcing. Other studies using POLDER show less strict requirements on radiometric accuracy, between 1 – 3% (Fougnie et al., 2007). Meeting these metrics has required further investigation of the wide FOV characteristics of HARP2 compared to prior iterations as HARP2 is a pushbroom scanner where the FOV characteristics directly impact the measurements at differing view angles.

       The AirHARP Instrument design is well documented in (Mcbride et al., 2024) and in terms of its broad characteristics,
HARP2 is much the same. The instrument utilizes a wide FOV lens to capture the ground target in a continuous scan. Sequential images are sliced apart into *view sectors* according to their shared view angle in the along-track direction. A spectral filter on the detector physically demarks the view sectors and isolates their spectral band which alternates according to a pre-defined pattern on the along-track axis. Its internal polarization identifying optical element is a beam-splitting Phillips prism, designed to isolate scene polarization to 0°, 45° and 90° relative to the instrument's direction of flight in a division of amplitude system
design. The final images produced are pushbrooms, which are a combination of the three polarization channels converted to the Stokes parameters according to a characteristic linear equation described in (Mcbride et al., 2024).

       Like AirHARP, HARP2 possesses 4 spectral channels centered at 440, 550, 665, and 865 nm. HARP2 uses the same number of red (665 nm) view sectors (60) but reduces the number in the remaining channels to no more than 10. The view sectors correspond to the angular scans of the instrument, and the red channel provides a high angular resolution for
measurements of the polarized cloudbow (McBride et al., 2020), whereas it has been shown for measurements of aerosol properties that 10 viewing angles is already more than enough to be sufficient (Wu et al., 2015). HARP2 also has been improved over prior iterations by including a pair of shutters which provide on-orbit dark and solar diffuse captures for system degradation monitoring. On-orbit, HARP2 is temperature controlled by balancing internal heaters and a dedicated space-facing radiator. During commissioning, HARP2 demonstrated thermal control at -13 ± 0.2°C for the three CCDs. After first light on
April 11, 2024, the thermal setpoint was changed to -18°C to further reduce dark current noise. All HARP2 components were previously validated in space in the HARP technical demonstration CubeSat (J. V. Martins et al., 2018), which performed 60+ test captures over a two-year period from 2020 to 2022 before orbital decay finally caused it to enter the atmosphere.

       In September/October of 2022, HARP2 underwent pre-launch calibration at the NASA Goddard Space Flight Center (GSFC) Code 618 Calibration Laboratory. We initially focused HARP calibration at the center of the instrument FOV and
90 inferred that these coefficients could be spread across the entire CCD via a pixel response normalization (flat-field). For HARP2, this telecentric method was challenged and the calibration activity focused on taking data at multiple points across the FOV for validation of the method. The results of these tests will be important for any future instruments using a wide FOV front lens and possessing polarization sensitivity. In Section 2 broad details of the experimental setup are provided, describing what tests were done and how they were accomplished. Section 3 presents the results of these tests with emphasis on the
95 variability of measurements across the FOV. Finally, Section 4 summarizes these results and provides discussions on recommendations for future MAP characterization activities and how our results may inform the long-term system monitoring of HARP2 on PACE.

## 2 Experimental Setup

HARP2 calibration began in September 2022, at NASA Goddard Space Flight Center (GSFC) Code 618 Calibration Laboratory. The experimental setup (see Figure 1) included the GSFC Grande integrating sphere (Kelley et al., 2023), a wire-grid polarizer on a stepper rotator motor, and HARP2 itself on a dual-axis motorized mount controlled via the HARP2 instrument control software. Additionally, the smaller Venti integration sphere was also used in conjunction with the Goddard Laser for Absolute Measurement of Radiance (GLAMR) system for spectral characterization (Barsi et al., 2023) in a similar setup, lacking only the external polarizer. HARP2 operated at a constant, sub ambient temperature ($18°$ C) throughout all tests thanks to a dry purge cooling loop over the radiators.

Using the GLAMR system, the HARP2 spectral bands were characterized using a separate "9-Sector" scan pattern. During these tests, stabilization of the GLAMR laser wavelength and power by the operators informed the HARP2 control system when to begin acquisition. Therefore, the time window of acquisition was limited, and a smaller number of scan sectors was used than in tests with Grande. The pattern was designed to maximize the FOV coverage via use of 9 sectors. An example of the scan pattern, and the control angles used for it, are shown in Figure 2 and Table 1, respectively. This scan pattern had a duration of just under 90 seconds and was effective, in part, thanks to being able to get the instrument closer to the Venti sphere than to Grande. To cover the spectral response function (SRF, or sometimes called the Relative Spectral Response, RSR) of HARP2, GLAMR first stepped through wavelengths at a coarse 5 nm resolution around the expected SRF for each HARP2 band and then performed a secondary pass at a much finer 2 nm resolution when the bounds of each band were roughly determined. The HARP2 SRF thereby possesses a non-uniform wavelength coverage, but with sufficient resolution to properly characterize it. Characterization of the signal between the HARP2 bands was performed using even broader ( $>$ 5 nm) steps.

Now, consider Grande, which possesses 9 internal incandescent lamps which can be turned on independently to linearly vary the light level. One lamp contains a variable attenuator which allows for modulation of that lamp's output illumination. The interior of Grande is coated with a broad-spectrum scattering coating that ensures light leaving the 25.4 cm aperture is spatially uniform and unpolarized. Experiments with similar integrating spheres show depolarization down to $<$ 0.01 absolute DoLP (Ding et al., 2011; S. C. McClain et al., 1995). The output has a radiometric accuracy of approximately 1% (Kelley et al., 2023), with slight variation across the wavelength range and at low illumination levels. For polarimetric calibration activities, two lamp levels were used: 3 fully illuminated lamps, and 7. This was intended to counteract the steep change in intensity from the blue wavelength range to the near-infrared (NIR) inherent in the Grande incandescent lamps' output. The "3-lamps" level ensures that light in the HARP2 red and NIR channels will not saturate, but as a result the blue and green channels will have a low signal to noise ratio. The "7-lamp" level improves the blue and green signal while ignoring possible saturation in red and NIR.

For both lamp levels, the wire-grid "generating" polarizer was sequentially stepped through rotations at 20° intervals from 0° (relative to the instrument cross-track axis) to 360° (inclusive) resulting in 19 different states of polarization measured. The generating polarizer was a Moxtek 20 cm PPL04A custom coated wire-grid polarizer with a contrast ratio of 1000 at the

center of the HARP2 blue band (440 nm), and monotonically increasing until around 800 nm with a contrast ratio of about 8000, whereupon it slowly begins decreasing through the HARP2 NIR band (865 nm), remaining over 7000. Note also that the generating polarizer was tilted around an axis perpendicular to the instrument optical axis (parallel to the cross-track direction of HARP2) by approximately 13° to avoid back-reflection into the instrument. We know from ray-tracing simulation that this tilt imparts an uncertainty in the generating polarizer rotation angle of up to 1 degree, depending on angular position but this is not yet accounted for in our analysis while the simulation is improved. At each measurement step, the internal HARP2 detectors took images one after another in a short 5 – 10 second interval at full resolution. For each step of the polarizer, motorized controls also performed a roll and yaw operation to view Grande at up to 26 different sectors of the FOV. The pattern of this "26-Sector" scan was determined in such a way as to minimize the movement between positions while covering as much of the FOV as possible in an appropriate time interval. The specific angles of each position in the scan are listed in Table 2, while a composite image of all scans from a single dataset show the relative pattern in the detector in Figure 2, including superimposed numerals depicting the scan order index starting at 0 and ending at 26. An additional "27th sector" was taken at the end of each scan at the center position, the same as sector 26, but here both shutters were sequentially actuated without further movement of the dual-axis mount between acquisitions, and the integration time maximized for characterization of the instrument diffuser and dark operational modes. The "26-Sector" scan, as it will be referred to from now on, was also used for tests of the bare Grande sphere with no generating polarizer in place. These were done at all 9 Grande lamp-levels and at an additional 4-lamp levels where the lamp with the variable attenuator was set to 50%. This produced interstitial "half-lamp" levels, though the radiances at these levels were not exactly halfway between the corresponding whole lamp-levels taken with the aperture fully open and therefore the half (or 0.5) designation is merely colloquial. Therefore, the radiometric tests were done at 13 total lamp levels listed as: 0.5, 1.0, 1.5, 2.0, 2.5, 3.0, 3.5, 4.0, 5.0, 6.0, 7.0, 8.0, 9.0.

## 3 Experiment Results

The initial pipeline for HARP data involves the correction from raw counts ($c_{raw}$) to corrected counts ($c_{corr}$) via the application of several standard transformations defined as:

$$C_{corr} = \frac{NLC(c_{raw} - D)}{\frac{1}{N_F}NLC(F_{raw} - D)}, \qquad (1)$$

where $F_{raw}$ corresponds to the flat-field image data from the HARP2 diffuser, $D$ corresponds to the expected dark frame data in counts units for same, $N_F$ is the normalization of the flat-field signal acquired by taking an average at the center of the FOV for each band (therefore each HARP2 spectral band has its own $N_F$ parameter, per detector), and the function defined via $NLC$ is a transformation to ensure the linearity of the counts data to increasing radiance (more on this in Section 3.2). This correction,

Equation 1, occurs for each pixel in a given image, though the entire denominator can be pre-calculated and treated as a scalar multiplier image colloquially referred to simply as the "flat-field" whose role is to ensure all detector pixels have similar counts levels for the same external illumination.

For all calibration activities, a pixel was selected for evaluation and averaged with its surrounding pixels to reduce uncertainty in the measurement (like done to acquire $N_F$). These "super-pixels" were a simple arithmetic mean of a selected pixel and a window surrounding it that in total contained 5 pixels along-track and 19 pixels cross-track; note that these super-pixels were not square due to the spectral stripe filter on the HARP2 detector having discontinuities in the along-track direction as a part of the pushbroom design functionality. Therefore, extensions in the cross-track direction were preferred to improve signal to noise of the mean. The uncertainty ($\sigma$) in the counts measurement of a given super pixel containing $n$ subpixels was found via standard error propagation. Before this calculation, we assume that the uncertainty in any given pixel is directly proportional to the Poisson noise of the distribution of electron capture events on the CCD. In cases where the process being applied to the data may have a non-continuous derivative such as the optimization of the calibration matrix coefficients, we instead use a Monte Carlo methodology for error propagation, where the input uncertainties are added to the input data as random noise and the non-continuous process repeatedly performed with changing noise values from a random number generator like that done in (Ramos & Collados, 2008). The expected result is the average of all these processes, whereas the uncertainty in the process can be found via the standard deviation of the different results about that mean. For these cases the results were typically repeated for 1000 Monte Carlo iterations. Note additionally that the entire calibration process has non-linear dependencies (e.g. the SRF utilizes measurements from each detector, combined by the polarized calibration matrix, but the calibration matrix itself requires a system SRF for optimization, for example. This is a circular dependence.). Therefore, the final coefficients are determined iteratively, with preliminary fits informing the final fits until the solution stabilized. Also note that the uncertainties of the polarization calibration matrix were evaluated only for the diagonal elements of the uncertainty matrix, ignoring covariance terms between the different elements. This was done for the sake of simplicity in the final analysis of the wide-FOV analysis done in this paper. A more rigorous treatment of HARP2 total instrument uncertainty will consider these.

Finally, in all cases the dark frame ($D$) was found via a temperature-stable average of 10 images taken in the lab with the calibration sources turned off. This was done to capture a variety of external light sources that may have otherwise biased the data, such as the glow of computer screens or light leakage from nearby lab spaces. The flat-field data used was from the normalization of the diffuser shutter data taken at the brightest Grande lamp level with maximum integration time in addition to bare images of Grande taken by the instrument at calibration integration times and a large number of tip/tilt mount positions

## 3.1 Spectral response Function

To characterize the spectral response of the system, HARP2 operated in conjunction with GLAMR for about 2 weeks, over half the total calibration time allotted. During these tests, 233 valid GLAMR scans were performed, each scan producing

1 image per HARP2 detector for each of the 9 scan positions (excluding tests done with the diffuser or dark shutter). This resulted in about 50 gigabytes of image data, excluding backup/redundant images and instrument metadata. A combination of a brute force search algorithm and hard-coded, operator input was used to identify suitable locations for super-pixel aggregation during this calibration activity. A software glitch in the HARP2 control software fixed after the GLAMR calibration activity caused image acquisition to occur during the movements of the dual-axis mount. In these cases the circular target was located in a slightly different position of the FOV, meaning super-pixel locations had to be adjusted accordingly and a brute force search of illumination gradients was used for these adjustments due to the amount of image data preventing a human operator being able to manually make the adjustments necessary.

The first major observation from the SRF tests with GLAMR showed a wide FOV effect visible in Figure 2 as a non-uniformity of the sphere illumination across the target regardless of its FOV position. Figure 2 is made from a composite of the scan pattern in the red-band wavelength range showing this effect. This anomaly was determined to be the result of the HARP2 wide FOV being able to "see" the location of the first laser bounce inside the sphere which was confirmed to be a known limitation by the GLAMR operational team. GLAMR inputs its laser light into the Venti integrating sphere via a fibreoptic cable positioned such that an instrument with a narrower (e.g. $< 10°$), FOV looking directly into the sphere will only see the result of secondary bounces of the laser, producing a uniform illumination. Only HARP2's unique FOV revealed the extent of this effect. We observed that the first bounce signal could be 50% stronger (or more) than the signal at the center of the target which was what is used for the actual calibration of any instrument, HARP2 included. Accurate radiometric calibration using GLAMR would need to ensure masking/normalization of this effect for calibration of wide FOV instruments. While it may be possible to mechanically adjust GLAMR to correct this effect (via pointing of the input fiberoptic or a diffuser in the optical path), doing so without affecting the radiometric accuracy of the system is non-trivial and not something supported during this calibration activity as the vast majority of other instruments are unaffected. For our needs in producing the HARP2 SRF, the binned super-pixel used to generate the response curve was simply taken to be as close to the center of the sphere aperture as possible for each scan position. Figure 3 shows the response curve of super-pixel data for each HARP2 band at all 9 sectors overlaid atop one another, as well as a scatter of the full-width-half-maximum (FWHM) bandwidth of each band and the center wavelength, as labelled by scan sector. From this we see that the bandwidth and bandcenter are very stable across the FOV, with precision well within 1% of their mean value. Different sectors vary in terms of the absolute magnitude of each SRF, showing that there is more structure than can be corrected by the flat-field, but the absolute magnitude of the SRF does not matter for the final product, as can be inferred from the name "relative spectral response." Although, in the cases of cross-band contamination, the relative magnitude between the bands *is* important.

The final SRF for HARP2 (shown in Figure 4) is an average of all sectors. For a given wavelength, the average response from all sectors was found, and the uncertainty determined by the standard deviation of same; the direct error propagation, thanks to super-pixel binning, was negligible compared to this variability and therefore the standard deviation metric is a good representation of the SRF uncertainty. The averaging process was performed only where at least 3 sectors had valid data, where valid data was defined as that data which had a signal to noise ratio of $> 6$ to avoid mislabelling of variability

in the dark signal as an actual detector response to light. This primarily ruled out cases done at wavelengths outside of the primary HARP2 bands, as expected, or between-band contamination within the HARP2 bands. Figure 4 clearly shows that some signal remains for the HARP2 blue-band response to light in the NIR and red wavelength ranges. This contamination is on average $< 0.5\%$, as normalized to the maximum signal in the HARP2 band where that wavelength produces the strongest response (i.e. the response at 865 nm in the HARP2 blue band is normalized by the response of 865 nm in the HARP2 NIR band). Also recall from the Section 3 header that the SRF is a "system-wide" calibration which requires the averaging of multiple sensors via the system polarization calibration matrix (See Section 3.3).

## 3.2 Radiometric Calibration

The radiometric calibration for HARP2 involved the stepping of the Grande lamps through increasing illumination levels and acquiring pictures with the 26-Sector scan at each lamp level. By convolving the GLAMR determined SRF with the provided Grande spectrum, we can determine the band-averaged radiance level associated with image data in corrected counts units. Like the SRF data, the locations of each sector's super-pixel were hard-coded by the operator, but here with no need for a brute force search as the 26-Sector scan had much more stable pointing than the 9-Sector scan used for the SRF (due to a software glitch fixed in the lab after the GLAMR evaluation). The stability of pointing here refers primarily to the position in the FOV, as a pixel-coordinate, of the target circle and therefore the location of super-pixel aggregation. The radiometric data also supplemented the polarimetric data (Section 3.3), as well as provided a test for the linearity of the detector response to illumination.

Linearity is important to HARP2 because it is by linear combination of the 3 detectors that HARP2 measures polarization and therefore any non-linear response in the detectors breaks this critical assumption about how that data can be combined accurately. Additionally, linear measurements are scientifically useful for evaluation of small changes in illumination (high contrast). Previous iterations of HARP have used a parabolic Non-Linear Correction (NLC) (Mcbride et al., 2024), HARP2 did the same, but without a scalar offset term that would bias low counts data. The NLC should be an inherently non-spectral effect, differing only by the electrical properties of the CCD. Therefore, the red band was selected for fitting and the same coefficients used across all bands. The red band data achieves the full dynamic range of the detectors when observing the Grande sphere at all lamp levels with a significant number of points not saturated. Further, we assume the NLC to be the same for all pixels across the FOV, again because it is electronic/detector effect rather than an optical one; precursor analysis done with HARP2 supports this assumption, but further details are beyond this paper's scope.

To determine the NLC function appropriate for HARP2, we first identified a region of linear response (with respect to the true Grande radiances). As the detector response is expected to be linear above the dark counts level but well below saturation ($2^{14}$ counts) we chose to fit a line to the dark corrected counts between 0 and 5000, as a function of Grande radiance. The linear fit was then extrapolated across the dynamic range of the instrument and the difference between the linear fit and the high-counts measurements evaluated (Figure 5). The result for HARP2 showed the parabolic deviation from the expected linear response, leaving a transformation equation simply as:

$$NLC(c_D) = Ac_D^2 + Bc_D \qquad (2)$$

Where the fitting parameters $(A, B)$ were found via fitting of the true, dark-corrected counts ($c_d$) to the expected linear extrapolation. The expectation is that these parameters are found such that Equation 2 is approximately linear in the expected linear response range of $x \in [0,5000]$ and that $NLC(0) = 0$. For the HARP2 red band, we found these values for each sensor; the results are shown in Table 3 to demonstrate that the relative strength of the non-linear to linear coefficients.

Upon fitting the parameters for all three detectors, all measurements going forward were evaluated after being remapped by this function as according to Equation 1. Upon doing so, measuring the radiometric response was simply an extension of the work already done to fit Equation 2. A line was fit for the Grande radiance at all non-saturated lamp levels as a function of corrected counts (Equation 1). Figure 6 shows the linear fits for each band at the center of the FOV (sector 26); the slope of these lines ($\kappa_\lambda$) are the radiometric coefficients for HARP2, given numerically in Table 4. Figure 6 also shows the quality of the linear fit at varying radiance levels of Grande. Note that rather than evaluate the radiometric coefficient by detector, it was chosen to evaluate it at the system level using the calibration matrix (Section 3.2) the same as was done with the SRF. This simplifies later data processing for HARP2 by limiting the radiometric coefficient to a single number by spectral band ($\kappa_b, \kappa_g, \kappa_r, \kappa_n$), rather than having one for each detector, per band, which would result in 12 total coefficients.

## 3.3 Polarimetric Calibration

While the polarization calibration matrix affects all parameters which combine the 3 HARP2 detectors, most cases are only concerned with the system-wide intensity ($I$) corresponding to the first row of the polarimetric calibration matrix. The far more sensitive second and third row of the matrix correspond to the states of polarization denoted as the Stokes $Q$ and $U$ parameters. The difference between the angle of rotation of the generating polarizer around its optical axis to the same angle of the static internal polarizer at each of the HARP2 detectors follows *Malus' Law*, which is proportional to $\cos^2(\Delta\theta)$ (where $\Delta\theta$ is the difference in angular position). By fitting this expression, we can determine the true starting angle of the generating polarizer with respect to the HARP2 reference detector (detector 1, whose polarizer position is defined as 0° with respect to the normal of the HARP2 prism mount, or parallel to the along-track travel direction in the image plane) and from there the relative angles of the 3 internal polarizers.

The state of polarization for a rotating linear polarizer with a given angle $\theta$ has the Stokes vector defined in the Equation 3 below:

$$\mathbf{S} = \begin{pmatrix} I \\ Q \\ U \end{pmatrix} = \begin{pmatrix} 1 \\ -\cos(2\theta) \\ \sin(2\theta) \end{pmatrix}. \qquad (3)$$

In this formulation, we can define the HARP2 characteristic equation as:

$$OS = d, \qquad\qquad (4)$$

where the characteristic polarization matrix is defined as $C = O^{-1}$ and should approximately follow a Pickering form (Schott, 2009). In the formalism of (Iniesta & Collados, 2000), matrix $O$ is referred to as the modulation matrix. The data vector $d$ comes from the lab measurements of $c_{corr}$ for each detector and represents the "modulation cycle" of the calibration process. The matrix $S$, meanwhile, is the inferred Stokes representation of the modulation process after linear combination by $C$, the calibration matrix. Both $S$ and $d$ have dimensions of $m$x3, for $m$ measurements and 3 detectors. The modulation matrix, $O$, is a composite matrix where each row corresponds to the first three coefficients of the optical path Mueller matrix of each HARP2 detector (Iniesta & Collados, 2000; Mcbride et al., 2024). Therefore, the first element of the first and last row of $O$ correspond to the elements which modulate intensity for the two optical paths corresponding to the principal components of the HARP2 intensity channel (sensors 1 and 3), which possess orthogonal polarizers. The sum of these two elements can therefore be used to normalize $O$ and produce from its inverse a normalized polarization characteristic matrix for HARP2, $C$. The normalization of the matrix does not matter for the final calibration product, as everything is scalarly modified by the radiometric calibration coefficient (See Section 3.2), but normalizing the matrix makes it easier to judge if it follows a Pickering form or not and to understand the true impact of uncertainty on the radiometry.

Equation 4 follows the form of a linear matrix equation, and therefore can be solved by any number of standardized least-squares methods for over-determined problems. We chose to use the pseudo-inverse from Singular Value Decomposition (SVD) which is has been noted to be capable of finding the least squares solution to such an equation for all matrix elements (Iniesta & Collados, 2000). As noted in the introduction to Section 3, the full calibration process cannot be handled by standard error propagation because of it not having a smooth derivative and having circular dependencies which are iteratively minimized, and therefore we performed Monte Carlo repetition to judge the uncertainty in the fit of the calibration coefficients to uncertainties in the input measurements. These were found to be vanishingly small (on the order of $10^{-7}$) for 1000 iterations, meaning the solution was quite stable and the uncertainty in the final measurement is primarily determined by the uncertainty in the measurement itself rather than uncertainty of the fitted parameters of $C$. Additional analysis of the covariance terms of these parameters should still be performed in a complete HARP2 error model.

The data vectors in $d$ are created from both the polarimetric measurement data (which follows the Malus' Law form), as well as the radiometric measurement data (described in Section 3.2) concatenated together along the measurement dimension. In the ideal case, the polarimetric Stokes intensity would be ½ that of the Stokes radiometric intensity at the same lamp level due to the external linear polarizer meaning that the bare sphere radiance at the lamp level of the Malus' Law test is a natural normalization value; that is to say: the vector $S$, during fitting, is 1 at the measurement index corresponding to the bare Grande lamp level of interest. That lamp level's radiometric intensity can be found using the SRF, the same as is done in the radiometric calibration methodology. The Grande lamp level of interest is lamp level 7.0 for the blue and green bands, and

lamp level 3.0 for the red and NIR bands. Note though that there is one discontinuity in this methodology. The vector $\boldsymbol{d}$ now comes from two datasets concatenated together differing by the addition of the external generating polarizer, but a non-ideal generating polarizer will have a scalar transmissivity term on the right-hand side of Equation 4 only for the datapoints coming from the polarization data. Therefore, while solving for the SVD solution of Equation 4, we must also iteratively solve for a scalar transmissivity, $\tau$, which applies only to the data in $\boldsymbol{d}$ coming from the measurements of the generating polarizer (not the bare sphere) and which minimizes the mean absolute difference in $I$ for the same dataset (whose intensity should be stable as the lamp level is unchanging, only the state of polarization is being modulated). The result of this process can be seen in Figure 7, for the center of the FOV (Sector 26), where the polarized intensity is 0.418, 0.437, 0,427, and 0.418 for the blue, green, red, and NIR bands respectively. Figure 7 also shows that 50% of points were retained for evaluation of overfitting, which we found to not be significant.

This process was repeated for every sector in the 26-Sector scan, resulting in a different 9-element polarization matrix ($\boldsymbol{C}$) for each sector. Upon doing this, we noted that each sector varied in a systematic way inconsistent with the spatial distribution of the flatfield generated in earlier steps. We noticed that this variation roughly follows the expected angular response of our wide FOV barrel distortion and therefore must be a systematic deviation we need to correct for. In this case, "expected angular response" refers to the tangent of $dv/du$ where $v$ is the transformed coordinate of a rectilinear $y$ coordinate in the instrument FOV to barrel distortion and $u$ is a similar transformation of $x$. These transformations are typically even-ordered polynomials as a function of the radial distance from the instrument optical axis (Chellappa & Theodoridis, 2017) (see section 1.3.1.3).. There is correlation between this barrel distortion effect and the polarization state (the explicit values of the Stokes $Q$ and $U$ parameters), implying a polarization plane rotation, but a correlation also exists in DoLP. Therefore, in addition to possible polarization plane confusions, the effects of induced polarization may have to be considered and a robust full, polarimetric error covariance matrix will need to be developed alongside theoretical understanding of the laboratory reference frames and scattering properties within the instrument to provide a proper description of this effect. Here we attempt an empirical evaluation of the total effect for discussion in this publication. To characterize the polarization effect across all pixels, a 2-dimensional polynomial fit was used across the FOV in the form of a paraboloid:

$$f(x,y) = \alpha x^2 + \beta y^2 + \gamma xy + \eta x + \zeta y + \delta \qquad (5)$$

where $x, y$ correspondingly refer to the cross-track and along-track image coordinates, shifted such that the origin of the $x, y$ coordinate system lies at the optical center of HARP2. The free parameters ($\alpha, \beta, \gamma, \eta, \zeta, \delta$) are fitted from the data at all 26 sectors. Each of the 9 calibration matrix parameters gets its own fit of these parameters (which may be written as $f_{ij}$ according the $i^{th}$ and $j^{th}$ element of the polarized calibration matrix, $\boldsymbol{C}$), which are also independent by wavelength: 4 bands, 9 coefficients, 3 detectors combined gives 108 total coefficients to fully characterize the system. These coefficients in turn generate the calibration matrix coefficients at any point in the FOV.

To evaluate the performance of the paraboloid fit of the calibration matrix parameters, we used a comparison of DoLP across the FOV at the location of each sector's super-pixel. In Figure 8, we show first the mean difference $\left( MD_{DoLP} = \frac{1}{N} \Sigma_n^N (DoLP_n - DoLP_{ref}) \right)$ of each sector of the 26-Sector scan using the only fully polarized data ($N = 19$, for each rotation of the generating polarizer), where $DoLP_n$ is the measurement generated from the interpolated matrix at a given sector and $DoLP_{ref}$ is the measurement generated from the given calibration matrix found independently for that sector. For

comparison, we perform the same analysis where $DoLP_n$ represents the measurement generated using the center (Sector 26) calibration matrix, as was done for prior HARP iterations. (Mcbride et al., 2024) refers to this as the *telecentric technique*. We see a marked improvement using the interpolated matrix compared to the telecentric method, improving precision in the DoLP measurement by sometimes up to a factor of 10. Note that in Figure 8, the height of each bar represents the $MD_{DoLP}$ of all 19 polarimetric datapoints, whereas the error bars are representing the standard deviation of the MAD of each individual point.

The same process was done for the full vector, **d**, with the generating polarizer present (colloquially $DoLP = 1$, although in practice $DoLP$ was only approximately 1, within uncertainty) in Figure 8, and the intensity from the same dataset as well as that which did not have the generating polarizer (colloquially $DoLP = 0$), again within measurement uncertainty, in Figure 9.

## 4 Summary and Conclusions

       These HARP2 wide FOV calibration efforts reveal that the overall technique explored in (Mcbride et al., 2024)is not

entirely sufficient at all instantaneous FOVs. Whereas spectral effects (spectral response function and radiometric coefficients) do not have a spatial distribution across the HARP2 FOV, and therefore can be well captured by flat-field normalization. Polarization calibration, going forward, must therefore be handled differently. For wide FOV lenses, it is important to consider how the polarization calibration coefficients can systematically change across the FOV, and how we can characterize this without performing an explicit, pixel-by-pixel calibration which would produce massive data quantities. Our analysis shows

that there is a non-negligible degree of linear polarization (DoLP) effect, as shown in Figure 8 and 9, which can be vastly improved upon by fitting polarized calibration parameters to a continuous paraboloid function (Equation 5) across a set of broadly spaced FOV locations. At each location, the calibration procedure as done in (Mcbride et al., 2024) remains the same, and each element of the polarimetric calibration matrix then receives its own set of paraboloid fit coefficients to describe it's variation at all points in the FOV. Doing so is what will allow HARP2 to properly reach the PACE-desired 0.5% accuracy

requirement, as without it uncertainty in DoLP could be on the order of $1 - 5\%$, instead. l

       The HARP2 blue is of special note because it is expected that Grande radiance stability reduces for the HARP2 blue band (Kelley et al., 2023) and the HARP2 SRF indicates light leakage of long wavelengths into the blue band. For the latter, a simple correction can be applied which subtracts the dataset of the red and NIR bands from the dataset of the blue band after multiplication of the integrated SRF coefficient of approximately and 0.03% for the red band and 0.4% for the NIR band. The

effect, while notable for PACE data processing, does not significantly impact the comparison between the telecentric and paraboloid methodologies as described above.

        With respect to general considerations of the use of GLAMR for MAP calibration evaluation, this wide FOV analysis shows that the HARP2 SRF can safely use a telecentric evaluation technique. The greatest difficulty for wide FOV instruments seems to be from the first-bounce effect inside the GLAMR integrating sphere, though this is easily avoided via

masking/avoidance of the calibration data in this region. Far from the center of the integrating sphere aperture, signal enhancement of greater than 50% was observed with HARP2, as seen in Figure 2. Instruments with an FOV of greater than 10° ought to consider this effect in the future, but it is easily avoidable in data analysis.

        Finally, when it comes to the operational status of HARP2 currently aboard PACE, we must also address the existence of on-orbit corrections in the Level 1 data processing which are beyond the scope of this publication, but which follows from

further analysis of the above-described calibration data: An elevated background illumination present in each HARP2 detector. Currently it is believed this effect is the result of a combined back-reflection and defocusing of the raw Level 0 image data and is physically present in the instrument. Therefore, it is primarily handled by the HARP Image Processing Pipeline (HIPP), and actively applied in PACE data processing with HIPP version 3.10 or greater. This effect essentially correlates individual pixels with their neighbors (on the order of pixel distances of 100 at the resolution in this paper, though the effect decreases

rapidly with distance), increasing their measured illumination by approximately 5% depending on the surrounding pixel data. Because the HARP2 calibration dataset is uniformly illuminated in the regions of interest, correction of this effect does not significantly impact the polarimetric calibration coefficients, and their paraboloids. Instead, it is primarily of interest in the HARP2 Level 1 data products and more detail on the correction process will be given in the HIPP Algorithm Theoretical Basis Documentation. Intercomparisons with the PACE Ocean Color Instrument in the radiometric space indicate that HIPP currently

corrects well for this effect. Overall, our evaluation of the HARP2 calibration procedures has revealed that HARP2 is very much capable of meeting the accuracy requirements of PACE, with accuracy in DoLP approaching 0.5% (with variability depending on FOV position and band). The blue band should be treated with the most care, but even it shows typically < 1% uncertainty when the paraboloid methodology is applied over the original telecentric method. In the future, FOV-dependence must be considered for the calibration of wide-FOV polarimeters when it comes to their polarized dependence parameters.

Radiometric parameters, such as the SRF and radiometric coefficients, appear to be stable by comparison. For a system which is HARP-like, we show that a paraboloid of the second order is sufficient to account for the FOV-dependence on polarized calibration parameters. The HIPP algorithm continues to be updated to account for these effects and others. Future work should focus on understanding the theoretical basis for these polarization characteristics and broadening our uncertainty of the covariant uncertainty in our polarization metrics, beyond a simple uncertainty of DoLP. Further, more time should be given to

polarimetric calibration activities to expand their scope in the future. Understanding of HARP2 shall continue to iterate alongside theoretical understanding and optical simulations.

## Acknowledgements

The authors acknowledge the funding supports from the NASA FINESST (80NSSC21K1600) on behalf of Noah Sienkiewicz NASA PACE mission, the NASA ESTO InVest project, and the UMBC START award. Additionally, we acknowledge and thank the engineering and support staff at the Earth and Space Institute which have and continue to support all HARP iterations. Further thanks is given to Lorraine Remer for her contributions and insights on this project.

## Conflicts of Interest

The authors declare no conflicts of interest.

## Acknowledgements

HARP2 calibration datasets are available in netCDF format via a request to the corresponding author.

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

Figures:

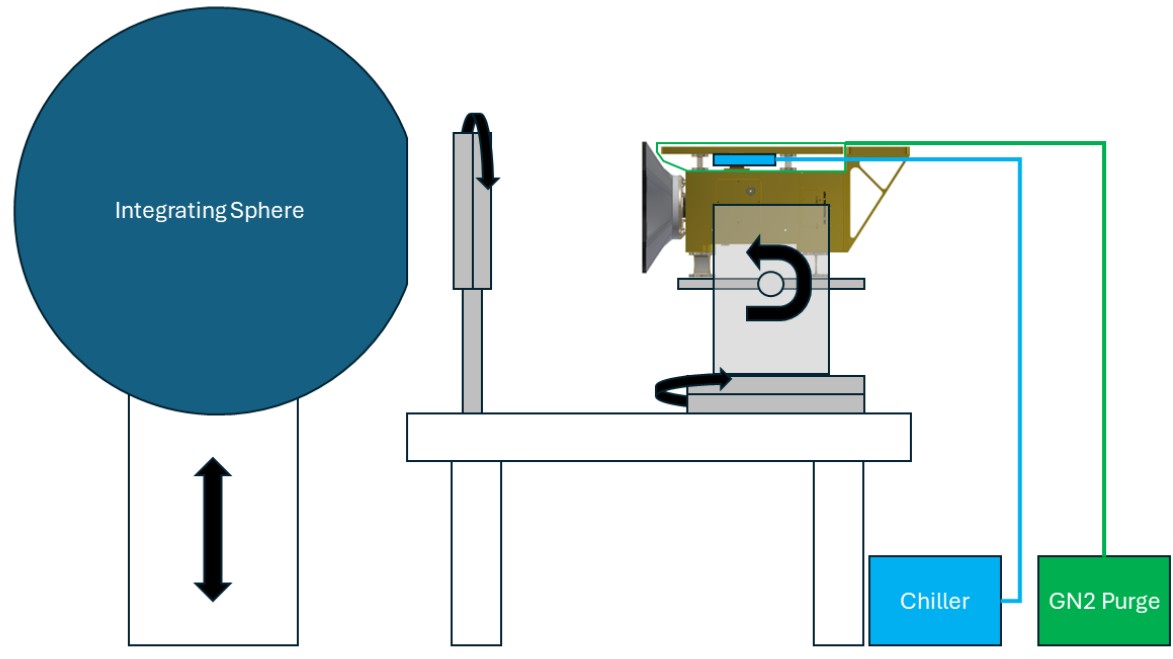

**Figure 1:** *Schematic of the HARP2 calibration setup showing HARP2 on the dual-axis yaw/pitch motorized mount, the Grande integrating sphere, mounted generating polarizer with stepper rotator, and the temperature control unit (Consisting of a cooler and a dry nitrogen purge line) tied to the HARP2 radiator. Black arrows indicate axis of movement for different parts of the system. The grey, vertical rectangle between sphere and instrument indicate the mount of the 20 cm diameter wire-grid generating polarizer. For Grande, the distance between sphere and instrument was 511 cm and for Venti the distance was 359 cm. The diameter of the Grande aperture is 25.4 cm, and Venti aperture dimeter is 20.3 cm.*

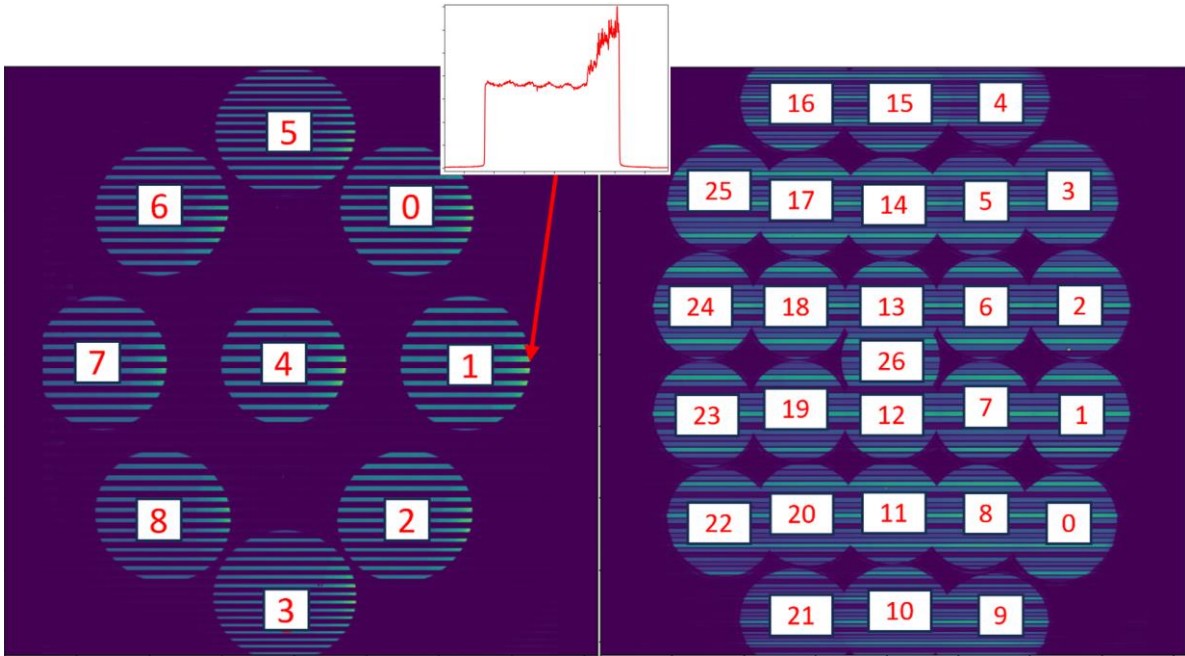

**Figure 2:** *(Right) Composite image of the 26-Sector scan of the HARP2 dual-axis mount, with numerical labels for the index ordering (starting at 0) for the polarimetric and radiometric calibrations. Circle targets are the Grande aperture as imaged by HARP at each scan position. (Left) The same photo using the 9-Sector scan utilized for the HARP2 spectral response characterization with similar numerical indexing. Here the circle targets are of the Venti sphere receiving its light from GLAMR. Horizontal lines in both images indicate the HARP2 spectral stripe filter. In the GLAMR (Left) image, only the red channel lines are illuminated due to the input of only red light coming from GLAMR, while Grande (Right) illuminates all spectral channels at the same time but with different brightness levels. Additionally, note that in GLAMR image data (Left), each circle target possesses a cross-track (left-to-right) gradient due to the first bounce of the GLAMR input laser. A cross-cut image has been superimposed at the location indicated by the red-arrow to show the magnitude of this effect.*

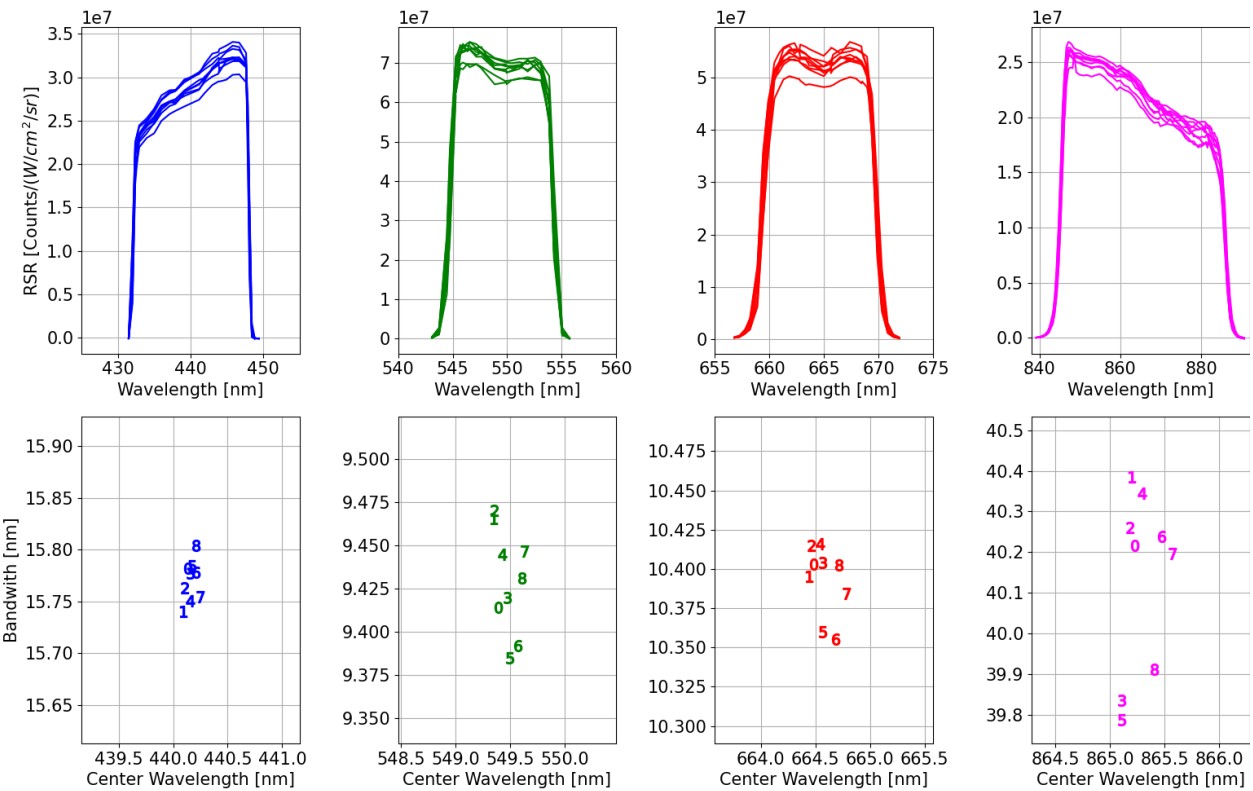

**Figure 3:** *(Top) From left to right the blue, green, red, and near-infrared plots of the unnormalized spectral response function as a function of GLAMR test wavelength. The multiple lines indicate the different scan positions used in the 9-Sector scan. (Bottom) In the same band order as above, a scatter of each spectral band's full-width-half-maximum bandwidth, and its center wavelength. Here the numerals indicate the index of the scan sector from which the data was taken (see Figure 2) with the bandwidth axis normalized to 1% of the mean value for each band.*

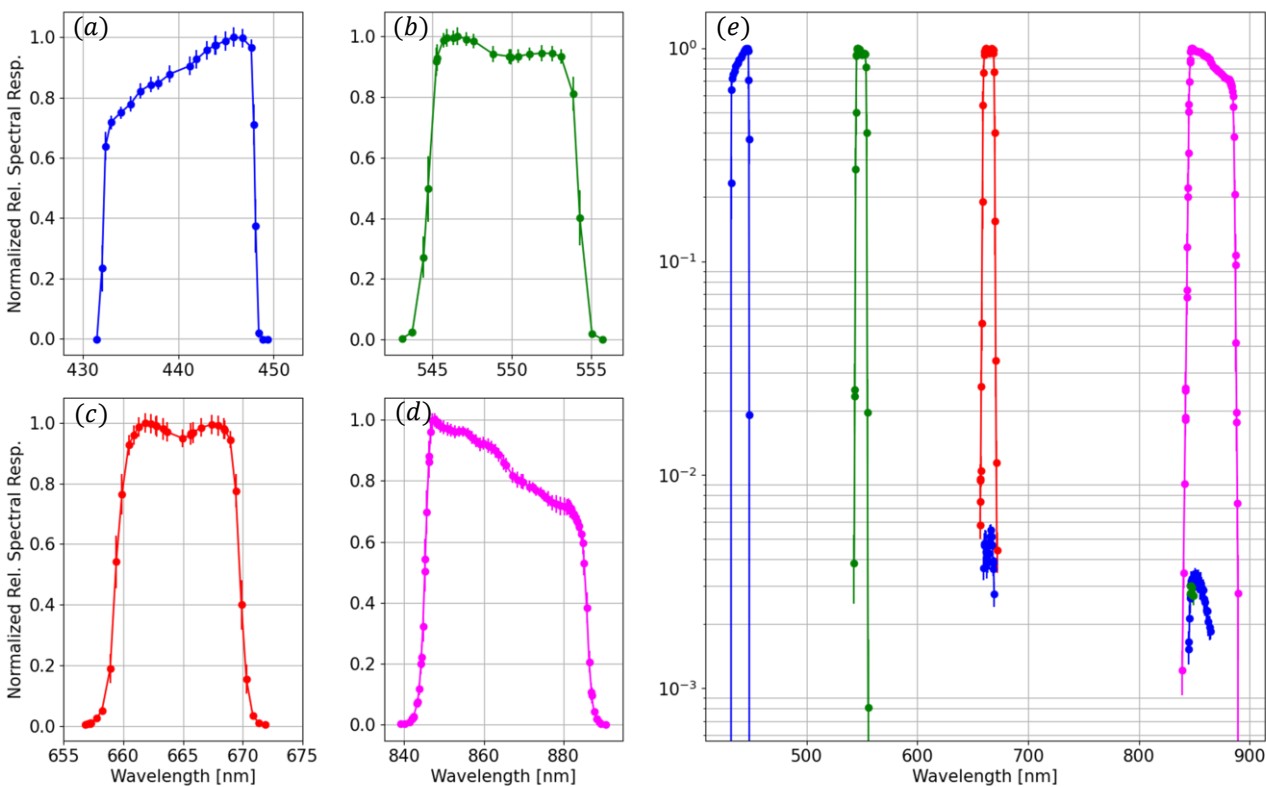

**Figure 4:** *(Left) The sector-averaged spectral response function of HARP2 with uncertainties for the blue (a), green (b), red (c), and near-infrared (d) bands. (Right, e) The full spectral coverage of each band response including the cross-band contamination visible in log-scale of the blue band response to light in the wavelength range of both the red and near-infrared (NIR) band. The green band also shows some contamination in just the NIR band. Band here refers to the response of the physical stripe filters on the HARP2 detectors.*

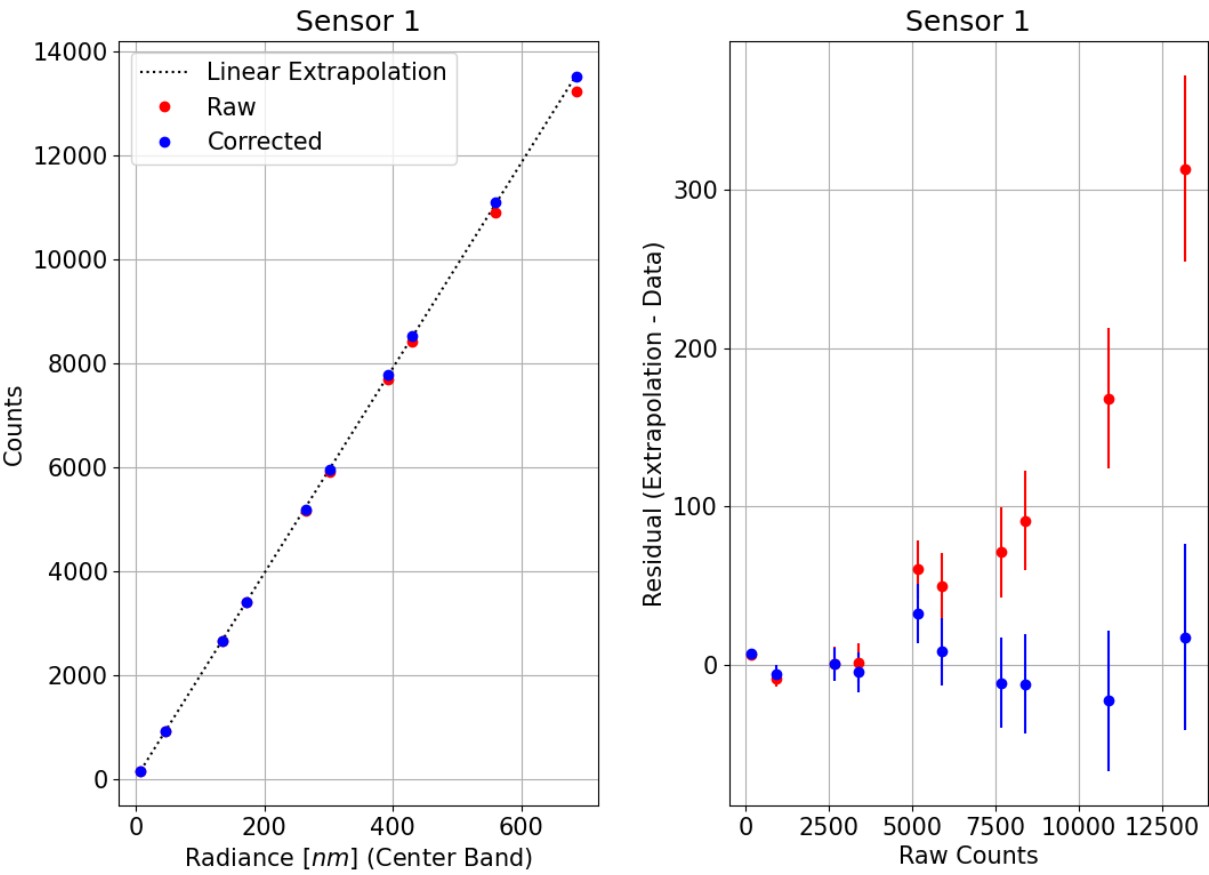

**Figure 5:** *(Left) The response of the HARP2 red band to increasing radiance. The raw data can be seen to deviate from a linear extrapolation from low counts data (0 to 5,000 counts) when nearing saturation (16,384 counts). The non-linear corrected data shows a much better adherence to the extrapolation. (Right) The residuals of the data both, raw and corrected, from the linear extrapolation with standard error.*

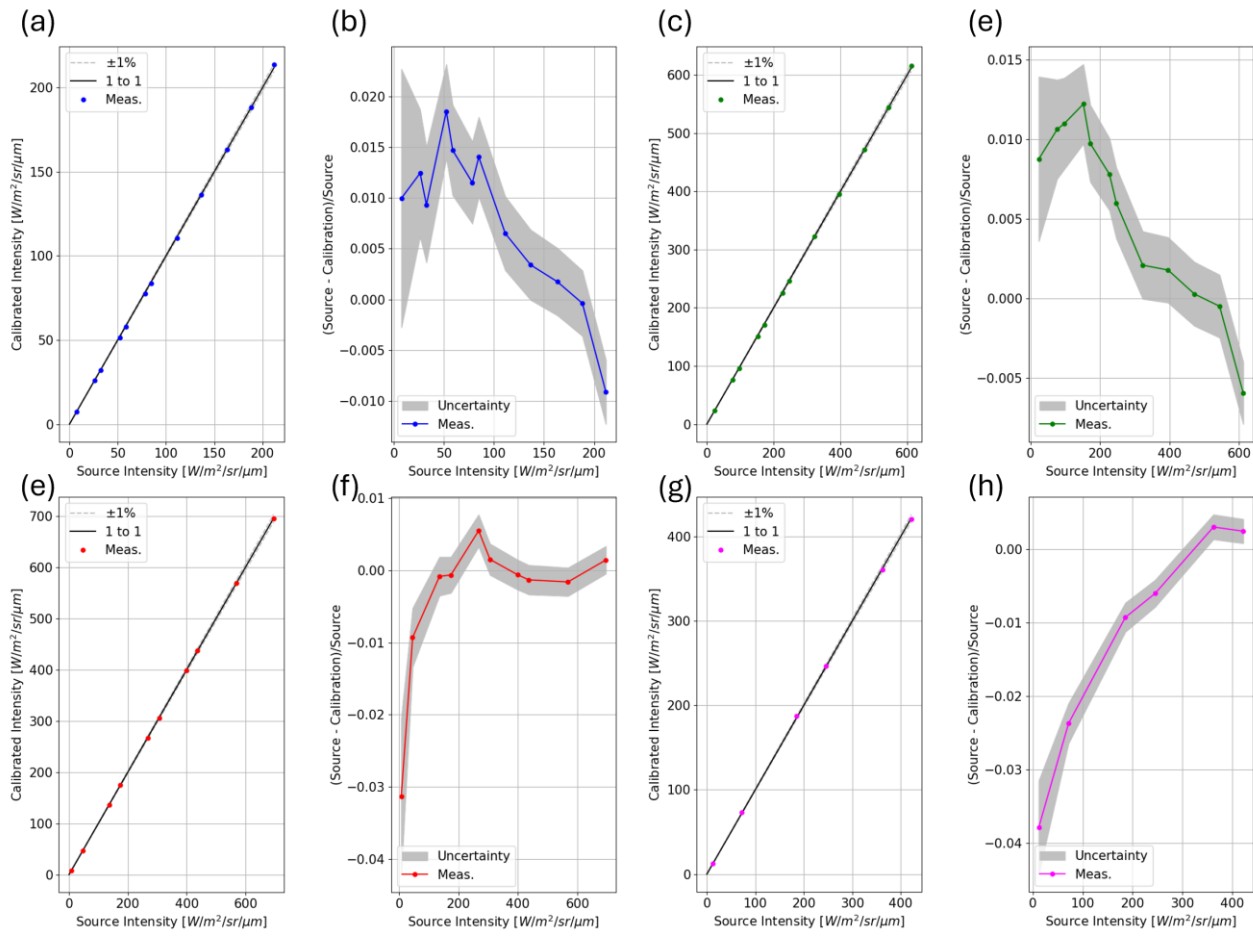

**Figure 6:** *For each HARP2 band, the linear fit of the radiometric intensity with respect to the Grande sphere intensity (a, c, e, g for blue, green, red, and near-infrared respectively), and the relative residual of the source (Grande) to the calibrated radiances of each fit (b, d, f, h in the same band order).*

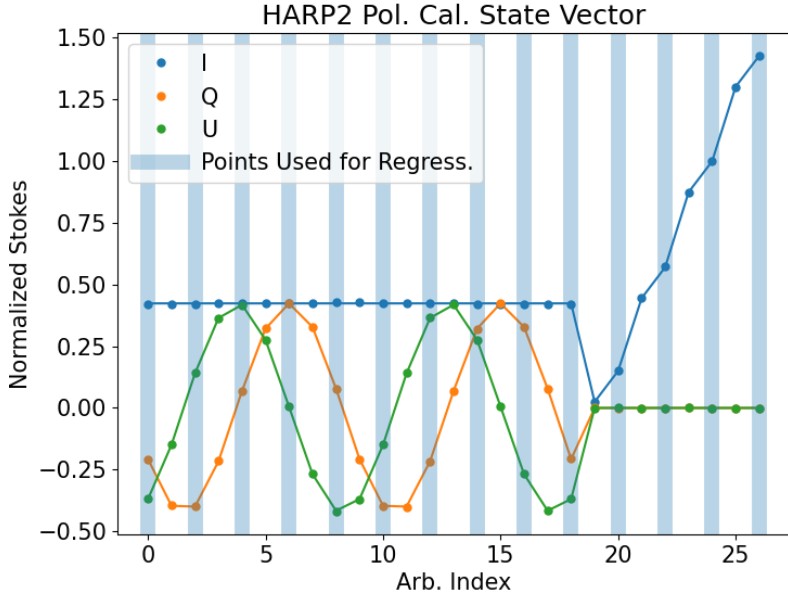

**Figure 7:** *Visualization of the HARP2 polarimetric calibration vector (**d**) over an arbitrary index for the red band. The polarimetric data (19 datapoints) consists of **20°** steps of the generating polarizer from **0°** to **360°** (endpoint inclusive) as well as all non-saturated data of the bare Grande sphere at varying lamp levels (here, for the red band, 8 datapoints). All datapoints are normalized according to the radiance level of the bare sphere at the same lamp level of the polarization data (here that is Grande with 3 lamps fully illuminated). The effect of the polarizer on the first 19 datapoints is to reduce by more than half of the intensity as compared to the bare sphere as well as provide changing Q and U measurements. The shaded blue, vertical lines highlight the 50% of points reserved for fitting of the calibration matrix, while the rest are retained for error analysis.*

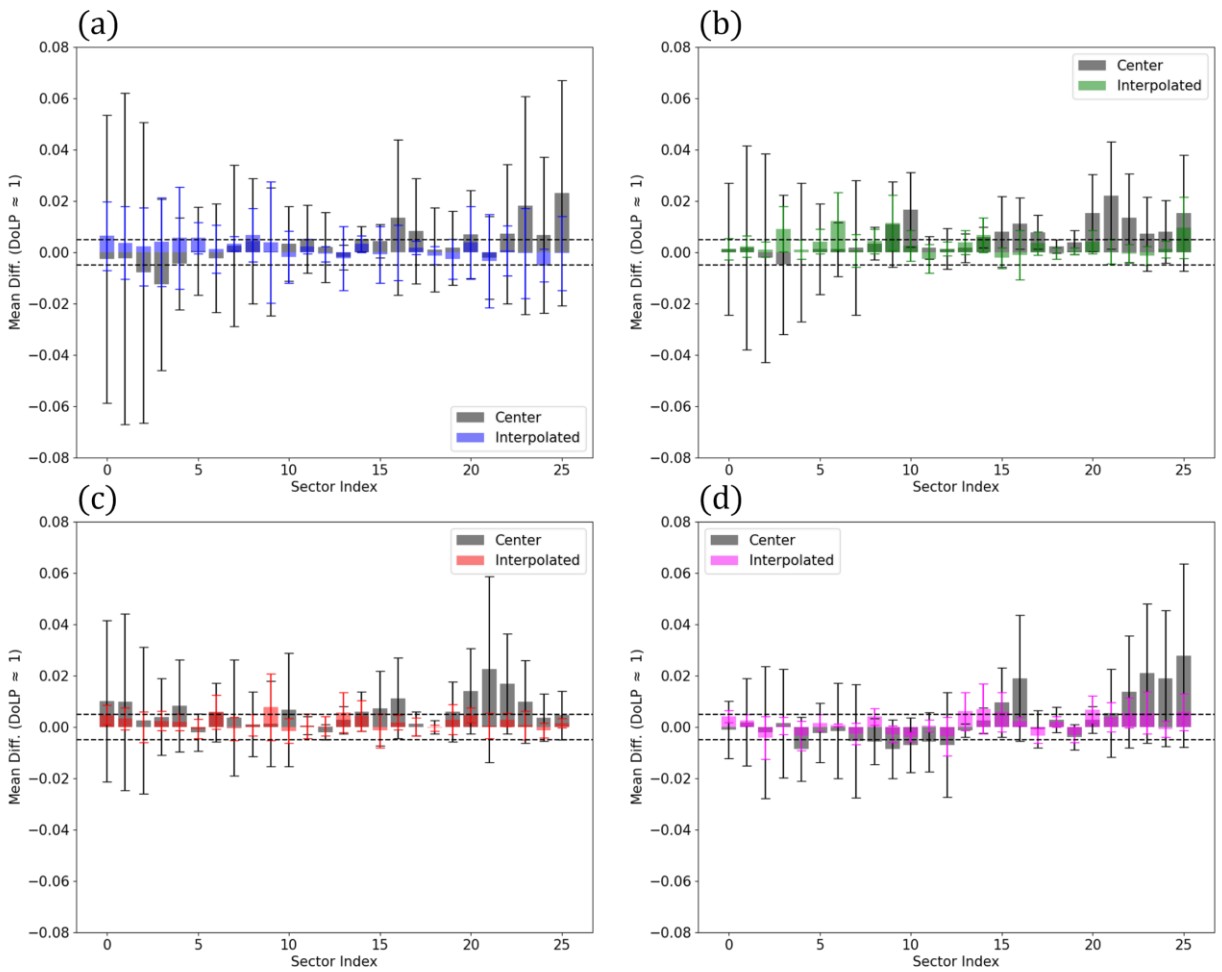

**Figure 8:** *Graphs indicating the degree of linear polarization calibration performance of the center FOV calibration matrix (grey) applied across all sectors in the 26-Sector scan (See Figure 2) and the performance of matrices generated from paraboloid fitting of the calibration matrix (colored by spectral band, or in order of blue, green, red, and near-infrared referred to also as a, b, c, d here). Bar heights indicate the mean difference of only the fully polarized data (DoLP approximately 1) as compared to the result from each sector's independent calibration matrix. Error bars indicate the standard deviation of same. Dotted lines indicate a $\pm 0.5$ uncertainty.*

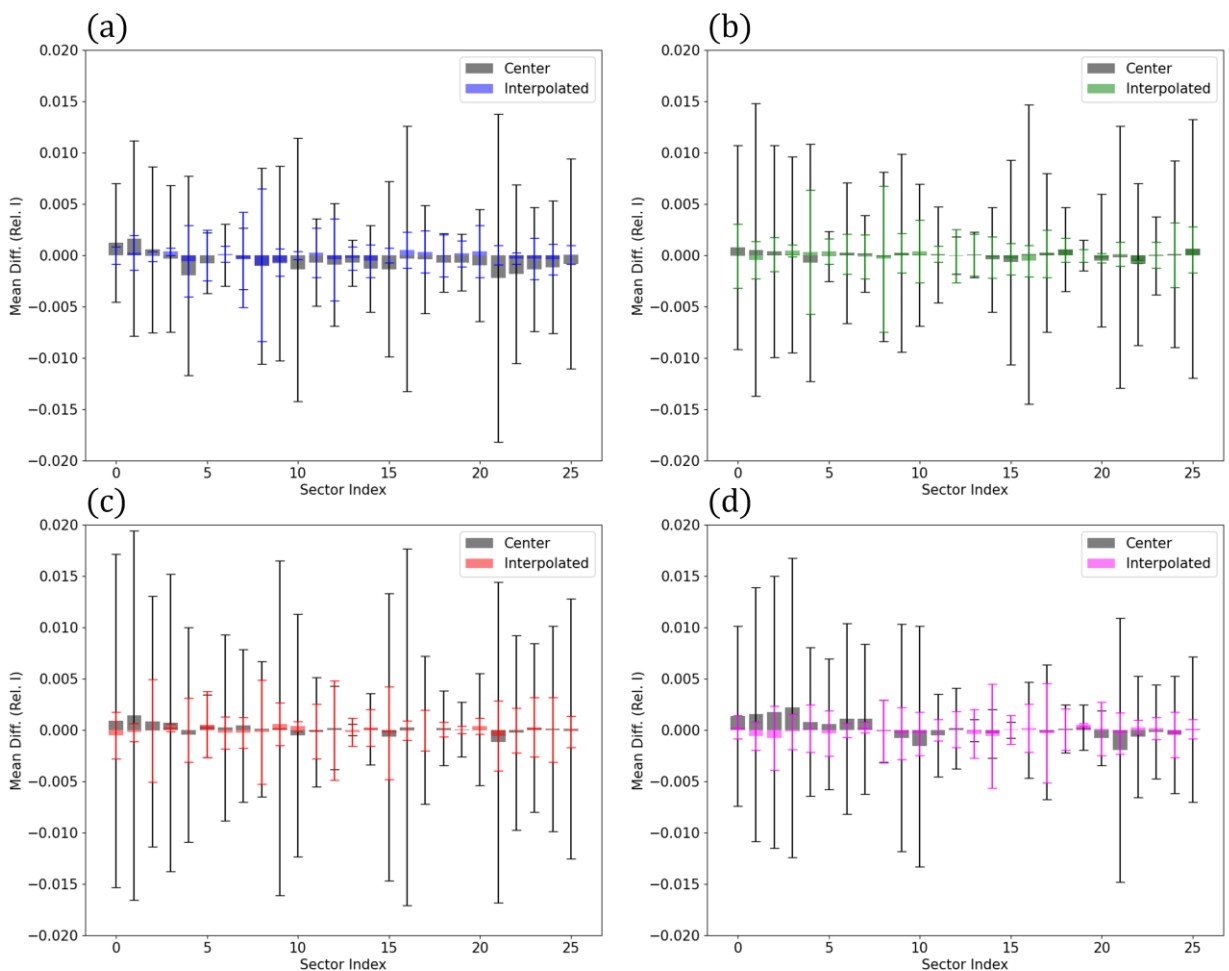

**Figure 9:** *Graphs indicating the degree of linear polarization calibration performance of the center FOV calibration matrix (grey) applied across all sectors in the 26-Sector scan (See Figure 2) and the performance of matrices generated from paraboloid fitting of the calibration matrix (colored by spectral band, or in order of blue, green, red, and near-infrared referred to also as a, b, c, d here). Bar heights indicate the mean difference of the fully polarized data (DoLP approximately 1), and unpolarized data (DoLP approximately 0) as compared to the result from each sector's independent calibration matrix. Error bars indicate the standard deviation of same.*

Tables:

**Table 1:** *List of the yaw/pitch positions of the HARP2 dual-axis mount throughout 9-Sector scan.*

| SCAN INDEX | PITCH (DEG.) | YAW (DEG.) |
|---|---|---|
| 0 | **28.476** | -25.717 |
| 1 | **0.000** | -35.124 |
| 2 | **28.476** | -25.717 |
| 3 | **-44.476** | 0.000 |
| 4 | **0.000** | 0.000 |
| 5 | **44.476** | 0.000 |
| 6 | **28.476** | 25.717 |
| 7 | **0.000** | 35.068 |
| 8 | **-28.476** | 25.717 |

**Table 2:** *List of the yaw/pitch positions of the HARP2 dual-axis mount throughout 26-Sector scan.*

| SCAN INDEX | PITCH (DEG.) | YAW (DEG.) |
|---|---|---|
| 0 | -29.670 | -36.790 |
| 1 | -9.910 | -36.790 |
| 2 | 9.985 | -36.790 |
| 3 | 29.610 | -36.790 |
| 4 | 49.370 | -25.420 |
| 5 | 29.610 | -18.420 |
| 6 | 9.850 | -18.420 |
| 7 | -9.910 | -18.420 |
| 8 | -29.670 | -18.420 |
| 9 | -49.290 | -25.420 |
| 10 | -49.290 | 0.000 |
| 11 | -29.670 | 0.000 |
| 12 | -9.910 | 0.000 |
| 13 | 9.850 | 0.000 |
| 14 | 29.610 | 0.000 |
| 15 | 49.370 | 0.000 |
| 16 | 49.370 | 24.420 |
| 17 | 29.610 | 18.420 |
| 18 | 9.850 | 18.420 |
| 19 | -9.910 | 18.420 |
| 20 | -28.670 | 18.420 |
| 21 | -49.290 | 25.420 |
| 22 | -29.670 | 36.790 |
| 23 | -9.910 | 36.790 |
| 24 | 9.850 | 36.790 |
| 25 | 29.610 | 36.790 |
| 26 | 0.000 | 0.000 |

**Table 3:** *Non-linear correction coefficients according to sensor.*

|  | SENSOR 1 | SENSOR 2 | SENSOR 3 |
|---|---|---|---|
| *A* | 2.104E−06 ± 2.5E−07 | 2.300E−06 ± 2.1E−07 | 2.183E−06 ± 1.642E−07 |
| *B* | 0.9946 ± 2.6E−03 | 0.9912 ± 2.2E−03 | 0.9925 ± 1.8E−03 |

**Table 4:** *Radiometric coefficients by spectral band.*

| | RAD. COEFF. ($W/m^2/sr/\mu m$) |
|---|---|
| BLUE | 3.294E−02 ± 1.2E−04 |
| GREEN | 2.275E−02 ± 1.2E−04 |
| RED | 2.670E−02 ± 1.3E−04 |
| NIR | 1.743E−02 ± 5.0E−05 |