# Peer review of "HARP2 Pre-Launch Calibration: Dealing with polarization effects of a Wide Field of View"

_EGUsphere, 2024_

## Author Response (AR1)

**Reviewer #1**

Q: It remains unclear what the status of this paper will be in the context of the complete description of the polarimetric performance and calibration of HARP2. In particular, its relation to the McBride et al. (2023-2024) paper should be specified. But neither paper provides a complete overview of all contributors to the overall polarimetric uncertainty. Will there be one final paper that will provide a complete overview? For this paper, at the very least an overview of relevant instrumental polarization effects and other systematics (as a function of incident angle across the field of field; see below) should be provided, and from this, the pertinent effects should be identified.

R: **This paper serves as an expansion of the ideas in Mcbride et al (2023-2024), which provides an overview of error systematics for the HARP sensors (e.g. dark uncertainty, flat-fielding … etc). What McBride et al. does not characterize is how the more complex systematics change across the FOV. This paper provides the first attempt to do so using empirical data from the HARP2 calibration. Because the results of this analysis proved to be significantly different than expected, additional work needs to be done to finalize a complete overview of instrument uncertainty via simulation and theoretical review. The closing statement in the conclusion has been updated to reflect this.**

Q: The final results of polarimetric error are only discussed in a hand-waving fashion, and not interpreted at all. There is an allusion to "barrel distortion" without even the start of a physical description of how this effect would cause the polarization effects. To be accepted as a publication in this journal, at least a qualitative interpretation of the results is required.

R: **In response to another comment, an in-text citation and expanded description of the relationship between barrel distortion and the effects observed in HARP2 has been added at L339. It has also been noted in several comment responses that we are aware of several other effects that we must consider (such as incidence angle on the generating polarizer). Therefore, it is likely that a convolution of polarization plane changes from barrel distortion, external lab configurations, and instrumental effects such as induced polarization which are the culprit of the effects we empirically describe via the paraboloid model. Without additional lab measurements and/or a more advanced model, the convolution of how these effects are influencing one another prevents a greater qualitative description and it remains in the scope of an additional, third paper, as described in the response prior to this comment.**

Q: The requirement of 0.5% absolute polarimetric accuracy should be split up according to its constituent effects that are described by the 3x3 system matrix that relates the measured Stokes parameters [I,Q,U] to the "true" values. This matrix can be derived from the M and $M^{-1}$ matrices (one of which I understand to be the "polarized calibration matrix"?), as $M^{-1}*M - I$, with all possible errors and variations upon those incorporated in M. This 3x3 matrix can be normalized by the first element. The diagonal elements Q->Q and U->U ("scaling") then describe how well linear polarization is measured. Q<->U describes systematic rotations of the coordinate system (which

may not be that relevant). I->Q,U ("zero point") describes instrumentally induced polarization. And Q,U->I describes the influence of polarization on the radiometry. It appears that this manuscript predominantly deals with the first effect.

R: **The studies referenced here in the paragraph beginning at L55 which state a 0.5% uncertainty are heavily convoluted with differing aerosol retrieval scenarios (i.e. what retrieval method is used, what kind of surface and aerosol properties are being measured ... ) and therefore the error they use is usually a simple Gaussian uncertainty on polarized reflectance (directly proportional to DoLP uncertainty). If I am understanding your comment you are trying to understand how this connects back to the full theoretical treatment of a proper error covariance matrix. The references provided in a separate comment in your response are very enlightening in this regard and here is how I would relate the different regimes: To understand how instrument uncertainty relates to a wide variety of retrieval scenarios, retrieval scientists attempt to reduce the number of instrument uncertainty metrics which are varied in their studies because they are already varying a large number of other uncertainties in their retrieval schema (for ex. Low AOT vs high AOT, coarse vs fine aerosols, absorbing vs not-absorbing, land vs ocean ... ) rapidly increasing the complexity. Therefore, from the error covariance matrix you're describing, they are taking only the diagonal elements and convolving them together into a single error metric, through error propagation of the DoLP equation, to reduce complexity. This paper, as a response to those, is doing the same. In your notation, this means it is primarily attempting to provide quantification of the uncertainty of the I<->I, Q<->Q, U<->U uncertainties (the "scaling" uncertainties as you call them) together. I think that it is correct to say that this results primarily in a metric of instrument induced (or reduced) polarization, but it is not an isolation of this effect. The other uncertainties you describe are contributing here, but we lack sufficient experimental data to fully characterize them all independently and I believe this is why you indicate in another comment that this paper is insufficient to be considered a complete overview. As there is no explicit question in your comment, I hope that this will suffice for clarification. I would happily invite further discussion on this matter when it comes to experiment planning for future calibration activities to help pin down these individual covariance terms and our efforts to characterize them in simulations.**

Q: The main polarimetric systematic (which can be formulated as I->Q,U) is due to any differential effect between the three beams: transmission, sampling, optical aberration, imperfect non-linearity correction. The latter is alluded to in Sect. 3.2, but the propagation of the residuals to the polarimetry has not been performed.

R: **The primary science result of this study is to show if the interpolation of the calibration matrix elements by the paraboloid fit is an improvement over the use of a static calibration matrix used at all FOV locations upon being fitted only at the center of the FOV (the stated "telecentric" method). Finding the calibration matrix elements via fitting, requires fitting to raw data which already has these effects applied to it by virtue of being laboratory measurements from the instrument itself, rather than simulations. That is except for the non-linearity; which we prescribe during data analysis, and sampling; which differs in the field**

**than in the lab and must be handled separately regardless. Analysis done while excluding the non-linear correction does not change the conclusion of the validity of the paraboloid, so it has been excluded and expansion can be provided in later publications which will finalize the full, absolute uncertainty model.**

Q: At least the instrumentally induced polarization will likely vary significantly with incidence angle across the field of view. Is this effect considered? Have measurements of only the integrating sphere without a calibration polarizer been performed to map these effects?

R: **The HARP2 lens is itself telecentric and spherical, so the incidence angle at the lens front and at the detector interfaces should be minimal. To confirm this, yes, measurements of the bare sphere were taken and used to produce "flat-field" imagery. An in text correction at L190 has been added to clarify.**

**For additional detail, I note that differences in the paraboloid to telecentric comparison do not change significantly with the inclusion/exclusion of the flat-field indicating that our initial assumption is correct. This was true of both bare sphere flat-field, and diffuser induced flat-field, which primarily contribute to radiometric accuracy instead of induced polarization. We believe now that a more significant effect is actually the incidence angle of the light on the generating polarizer and we are in the process of preparing a correction for this via simulations.**

Q: Note that also Q,U can get lost to Stokes V due to birefringence in glass or reflections on metallic mirrors. Have such effects been considered?

R: **Effects of Stokes V have not been fully considered, no. From experiments on individual optical elements and simulations of them, we do not believe that the HARP system experiences significant birefringence, nor does it have any metallic mirrors. That said, instrument induced V does need to be further evaluated and from that determined whether or not additional corrections can be applied. This will have to come from simulations, or future lab activities with AirHARP2, HARP2's ground equivalent which we still have access to. This has briefly been noted as necessary in a revised version of the conclusion (see final paragraph).**

Q: Often, the systematics for Q and U can be considered equivalent. But this is not the case here, because of the sub-optimal three-beam splitting at angles [0,45,90] (instead of [0,60,120]). Because of this, the noise propagation to U is sqrt(2) worse. Moreover, Stokes U is highly susceptible to systematic effects in determining Stokes I from the [0,90] channels. Do you see any effects due to this "asymmetry"?

R: **We do have empirical measurements of the error in Q and U which do not appear to differ significantly. Certainly, we cannot confirm from our data an explicit sqrt(2) term in our data. As noted above, we understand now that there is a contribution to polarization plane uncertainty**

**due to the incidence angle of light on the generating polarizer. Until we understand and apply a correction for this effect, I do not think we will be able to see the result you describe. It might also be interesting to design a separate experiment where Stokes I is varied at a static polarization state to try and see this effect, which would remove uncertainties in polarization plane from obscuring the result. I would welcome more discussion on experimental design to measure this sort of effect.**

Q: l 118: I would hope that your integrating sphere is not reflective but scattering with high efficiency…

R: **In-text correction at L120: "The interior of Grande is coated with a broad-spectrum scattering coating …"**

Q: l 119: No beam of light is fully "unpolarized" even if only due to photon noise. Down to which level did you confirm polarization consistent with zero?

R: **In-text addition at L121 "Experiments with similar integrating spheres show depolarization .." including new citations.**

Q: 125: Please specify the wire-grid polarizer. Does it still have an extinction ratio >1000 in the blue band? And what is the effect of the 13 deg tilt? Perhaps in terms of 3D projection of the polarizer axis onto the beam?

R: **In-text addition at L131 "The generating polarizer was a Moxtek 20 cm PPL04A custom coated wire-grid polarizer with a contrast ratio of 1000 …" Also in-text addition at L135 "We know from ray-tracing simulation that this tilt imparts an uncertainty …"**

Q: Sect. 3.3 Please refer to https://opg.optica.org/ao/abstract.cfm?uri=ao-39-10-1637 https://opg.optica.org/ao/abstract.cfm?uri=ao-47-14-2541 for the formalism describing noise propagation and optimality of these matrices.

R: **Upon review of these papers, both were included as explicit references in several sections of the paper. From the first, the formalism/terms were directly worked into a rewrite of L290 through L316 and the Monte Carlo references from the second paper were used directly as it is a very similar method.**

Q: l 305: This "expected angular response" should really be explained! At least a qualitative comparison with the data should be part of this paper. Moreover, are you certain that there are no other effects that can explain the calibration measurements?

R: **Large in-text addition given for clarification beginning at L137 "In this case, "expected angular response" refers to … " including a textbook reference.**

Q: Fig 8-9: Because DoLP cannot be >1, the statistics in these plots are weird. Please elaborate on their relevance for measurements of scenes with DoLP<1.

R: **In-text addition at L365 to clarification. "The same process was done for the full vector … " I'm unsure if I'm misunderstanding your concern here. Fig 8-9 are uncertainties on measurements with approximately DOLP = 1, therefore they're not indications of measurements of DOLP > 1, only on how much variability around the expected DoLP we're seeing (which should be nearly 1, but are actually the result of using the explicit calibration matrix for that given FOV position, prior to paraboloid fitting). For scenes where DoLP < 1, we only have the case where approximately DoLP = 0 in the lab. Ideally, we'd also like to generate partial polarizations for 0 < DoLP < 1 but that was not possible during this calibration activity. To hopefully remedy this confusion, Figures 8 and 9 have been regenerated using an error metric which excludes the absolute value described near L365, converting mean-absolute difference to simply an average difference and updating the y-axis label to indicate "approximately". This is a less conservative metric, but removes the illusion that uncertainty was pushing real measurements of DoLP > 1. It also has the advantage of showing patterns in whether or not particular sectors are high or low biased by the paraboloid.**

**Reviewer #2**

Q: My feeling is that the title of the contribution does not match its content very well. Instead of providing an overview of the calibration, I find that the paper focusses on the difference with the previous approach. Therefore, I freely suggest: "HARP2 Pre-Launch Calibration: Dealing with polarization effects of a Wide Field of View" or something along these lines.

R: **Happy take the suggested change as I believe it is indeed more descriptive.**

Q: Abstract, p. 1: While probably obvious, it is not clearly stated that the presented, improved approach using the parabolic fit of the polarized calibration parameters will effectively be used to process the incoming flight data. The authors should elaborate on how this will work in practice (implementation in L0-L1 processor).

R: **In-text addition to end of abstract L26 "... but the improved methodology over the telecentric method is currently being implemented in the HARP2 L1B calibration pipeline pending internal review ... " while we've confirmed that the interpolation methodology works well in the calibration data, HARP2 has a unique on-orbit binning scheme and therefore we must internally validate the code which is doing the binning of these calibration parameters and applying it at L1B. This process will be included in the final HARP2 data processing ATBD.**

Q: P. 5, line 135: It is said that an additional "27th sector" was taken at the end of each scan at the center position, with both shutters sequentially actuated. For all clarity: so these are two extra measurements (diffuser and dark), but for the same position of the turn-tilt stages, correct? This could be clearly stated, to avoid confusion.

R: **Correct, I've added in-text expansion for clarity at L146: "without further movement of the dual-axis mount between acquisitions"**

Q: P. 7, line 199: Cross-band contamination: the cross-band contamination is weak and the contamination is ruled out by selection, in the SRF determination. This can be done here because monochromatic light is used (GLAMR) and the bands are (intendedly) illuminated one by one. In white light illumination (radiometric calibration for instance), all bands are illuminated at once, so the cross-band contamination will play a role. Is this contamination negelected? Please comment.

R: **Initially this was neglected, yes. Upon reviewing on-orbit PACE data, we expect that that this is an error and it cannot be neglected. More on this correction will be handled in the data processing ATBD, but for now the discussion paragraph has been updated to reflect this. The primary results regarding performance of the paraboloid fit remain unchanged.**

Q: P. 7, line 218: "...brute force search as the 26-Sector scan had much more stable pointing than the 9-Sector scan used for the SRF." It is not clear to me what is meant here. Why is the pointing for

the 26-Sector scan more stable? Besides, pointing should not be critical since the instrument is looking into a sphere. Please clarify.

R: **Added in-text citations beginning at L198 and L241 expanding upon the brute force search. To expand in this comment details which are beyond the paper's scope. Essentially the software controller, which was meant to send commands serially, experienced a race condition which caused the image acquisition commands to occur during movement of the dual-axis mount. Because the integration time is short relative to movement speed and the target is uniform, the data is still valid, but the data processing was complicated in that data selection regions for super-pixel aggregation could no longer be hard-coded for the 9-Sector Scan. The 26-Scan occurred later in the calibration timeline, and therefore we had time to fix this bug before changing configuration from GLAMR to Grande. I include the note about the brute-force search at all because technically speaking, it means that the results of the 9-Sector SRF scan may not be necessarily in the exact same FOV positions throughout all wavelengths and sectors. Because the SRF is so consistent across FOV, this is mostly unimportant, but had the SRF had more FOV-variability this is something that must be considered.**

Q: P. 12, top: It is stated that the existence of an elevated background illumination signal in HARP2,

discovered after the launch of PACE, is briefly addressed. Where is this done? Only here in the conclusion? If included, this deserves more attention (a paragraph on it wouldn't be out of place).

R: **The elevated background signal is an important effect, but it is also primarily an active correction applied in the PACE L1 data processing (i.e. a secondary correction applied alongside Equation 1) which involves manipulation of every L0 image as they are processed combined with coefficients determined from the here described calibration dataset. For the sake of completeness in a discussion of the calibration, and considerations for wide FOV instruments, it's mentioned here, but a full treatment is on the order of a dedicated discussion in the HARP2 L1 data processing ATBD currently being worked on. I do not think that an additional paragraph alone would be sufficient to properly describe the effect and may cause confusion and distraction from the primary result of the paraboloid methodology. Instead, I've chosen to expand the qualitative description of the effect in full rewrite of the conclusion. If further clarification is required, an additional figure might be more instructive.**

Q: P. 16, Figure 1: It would be instructive to indicate both the Grande and Venti spheres in this figures, with their respective output ports, along with the FoV of the instrument. This to give an idea of the proportions. Please also indicate the size of the polarizer used.

R: **Adding a second sphere for Figure 1 appeared quite cluttered on an attempt. Instead, I have updated the figure caption to provide quantitative values for the values requested: sphere aperture sizes and distances, and polarizer size. I've also updated the body text at L130 to include increased details about the polarizer used. Instrument FOV is stated numerically in**

the abstract and body text, while Figure 2 provides a better visualization of the portion of the instrument FOV filled by each sphere than an update on Figure 1.

Q: Technical corrections: ...

R: **In-text corrections for all technical notes listed (typos, acronyms ...)**